# Total Dietary Intake and Health Risks Associated with Exposure to Aflatoxin B_1_, Ochratoxin A and Fuminisins of Children in Lao Cai Province, Vietnam

**DOI:** 10.3390/toxins11110638

**Published:** 2019-11-02

**Authors:** Bui Thi Mai Huong, Le Danh Tuyen, Henry Madsen, Leon Brimer, Henrik Friis, Anders Dalsgaard

**Affiliations:** 1Department of Veterinary and Animal Disease, Faculty of Health and Medical Sciences, University of Copenhagen, DK- 1870 Frederiksberg C, DK-1870 Copenhagen, Denmark; buithimaihuong@dinhduong.org.vn (B.T.M.H.); hmad@sund.ku.dk (H.M.); lbr@sund.ku.dk (L.B.); 2National Institute of Nutrition, 48 Tang Bat Ho Street, Hanoi, Hanoi 100000, Vietnam; Ledanhtuyen@dinhduong.org.vn; 3Department of Nutrition, Exercise and Sports, Faculty of Sciences, University of Copenhagen, Frederiksberg C, DK-1958 Copenhagen, Denmark; hfr@life.ku.dk; 4School of Chemical and Biomedical Engineering, Nanyang Technological University, Singapore 639798, Singapore

**Keywords:** risk assessment, total diet study, aflatoxin B_1_, ochratoxin A, fumonisins, children, Vietnam

## Abstract

The health burden of foodborne mycotoxins is considerable, but particularly for children due to their lower detoxification capacity, rapid growth and high intake of food in proportion to their weight. Through a Total Dietary Study approach, the objective was to estimate the dietary exposure and health risk caused by mycotoxins for children under 5 years living in the Lao Cai province in northern Vietnam. A total of 40 composite food samples representing 1008 individual food samples were processed and analyzed by ELISA for aflatoxin B_1_, ochratoxin A and fumonisins. Results showed that dietary exposure to aflatoxin B_1_, ochratoxin A and total fumonisins were 118.7 ng/kgbw/day, 52.6 ng/kg bw/day and 1250.0 ng/kg bw/day, respectively. Using a prevalence of hepatitis of 1%, the risk of liver cancer related to exposure of aflatoxin B_1_ was 12.1 cases/100,000 individual/year. Age-adjusted margin of exposure (MOE) of renal cancer associated with ochratoxin A was 127, while MOE of liver cancer associated with fumonisins was 542. Antropometric data show that 50.4% (60/119) of children were stunted, i.e. height/length for age z-scores (HAZ) below –2, and 3.4% (4/119) of children were classified as wasted, i.e. weight for height z-scores (WHZ) below –2. A significant negative relationship between dietary exposure to individual or mixture of mycotoxins and growth of children was observed indicating that the high mycotoxin intake contributed to stunning in the children studied.

## 1. Introduction

Children are especially vulnerable to foodborne hazards due to their higher dietary exposure per kg body weight and differences in physiology compared to adults. Due to significant postnatal development of different organ systems during childhood, children up to four years of age are more sensitive to some neurotoxic, endocrine and immunological effects [1]. Dietary exposure to mycotoxins is associated with various health disorders and recognized as a major food safety hazard [2]. Among pathogenic mycotoxins, Aflatoxin B_1_, ochratoxin A and fumonisins are common and potent ones which can contaminate various types of foods [3]. The International Agency for Research of Cancer (IARC) [4,5] has classified aflatoxin B_1_ and mixtures of total aflatoxins into group 1: “Carcinogenic for humans”. Aflatoxins are documented causes of human liver cancer and impaired child growth, as well as an immunosuppressant [6]. The IARC has reported fumonisins as Group 2B as “Possible carcinogenic to humans” [5,7], based on evidence showing that fumonisins act as a promoter of liver and kidney tumors in rodents. Ochratoxin A has been evaluated to be carcinogenic in the kidney of some animal species, in addition to causing numerous other specific toxic effects, such as hepatotoxicity, teratogenicity and immune-suppressivity, in different animals [8,9,10]. Ochratoxin A is also classified into Group 2B as possibly carcinogenic to humans by the IARC [11].

Increased risk of liver cancer has been reported in people co- exposed to aflatoxins and hepatitis B virus (HBV) [3]. Thirty times higher risk of developing liver cancer was observed among individuals who experienced both exposures compared to those exposed to the mycotoxins only [12]. Vietnam is endemic for hepatitis B, with a prevalence of 7 to 24% among adults depending on age and geographic region. Of those, about 10 to 12% of pregnant women are chronically infected with hepatitis B. Hence, mother-to-child transmission is an important factor contributing to the high levels of chronic hepatitis B infection in Vietnam [13]. Newborn infants who become infected with hepatitis B virus show no symptoms, yet have a 90% chance of developing a chronic, life-long infection. By increasing the cover rate of vaccination, Vietnamese authorities expect to reduce the rate of chronic hepatitis B infection among children from 18% in 2003 to below 1% in 2017 [14].

Child malnutrition, including both energy- and nutrient deficiencies, is caused by multiple factors and are harmful to their health, growth, development, and burden of infectious diseases. Stunting remains common in Vietnam despite general economic development, particularly in areas with large populations of ethnic minority people such as the Central Highlands, Northern Midlands and mountainous regions [15]. About 25% of children younger than five years old in Vietnam are considered stunted [16]. The stunting rate among children in rural areas is twice as high as that in urban areas, while the level of stunting is approximately three times higher among Vietnamese children from the poorest households to which ethnic minority groups belong [15]. The Lao Cai province in the North West mountainous area of Vietnam is inhabited by 25 ethnic groups and has one of the highest prevalence’s of stunted children younger than five years of age countrywide. Based on nutrition profiles of the year 2014, 35% of the children younger than five years of age were stunted, 20% was underweight and wasting was seen among 6% of the children [16].

Chronic exposure to mycotoxins is increasingly seen as a threat to child health. Therefore, it is important to assess and predict the negative health implications of exposure to different mycotoxins. Exposure assessment, as one part of risk assessment, integrates mycotoxin contamination in food with consumption data and is used to identify which mycotoxins compromise food safety and health hazards [17]. Exposure data collected by so-called total dietary study (TDS) approaches consider and include all different foods consumed in the whole diet. Risk characterizations for the mycotoxins associated with cancer risk are available. Thus, the FAO/WHO Joint Expert Committee on Food Additives (JECFA) estimates the cancer risk for a certain population using the incidence of the hepatitis B virus (HBsAg+ individuals) and the carcinogenic potency of aflatoxins, which has been defined for HBV carriers and non-carriers [12]. The European Food Safety Authority (EFSA) and JECFA recommended to use the margin of exposure (MOE) approach to evaluate compounds that are both carcinogenic and genotoxic [18,19]. The MOE is the ratio between a toxicological threshold obtained from animal studies and the estimated human exposure [18]. A small margin of exposure suggests a higher risk than a larger margin of exposure. Hence, risk managers can use this information for priority setting [15].

Using the TDS approach, this study aimed at estimating the dietary exposure to aflatoxin B_1_, ochratoxin A and total fumonisins and the associated health risks among children younger than five years old in Lao Cai province, Vietnam.

## 2. Results and Discussion

### 2.1. Food and Nutrient Intake

Children were generally fed the same dishes as the rest of the family. Complementary foods were composed mainly of commodities from locally available food products (Table 1).

Common foods were rice, groundnuts, banana, beans, meat, powder milk, eggs and vegetables. The daily food-, energy- and nutrient intake are summarized in Table 2. The estimated daily mean energy intake was 870 (range 218–1713) kcal and mean protein intake was 28 (8–67) g. Daily intake of essential micronutrients such as vitamin A, iron and zinc were 99 (0–1044) mcg, 4.8 (1.0–9.5) mg and 3.7 (1.1–6.5) mg, respectively. The latter three intakes were lower than the national recommended daily intake [20].

Forty mothers and caregivers attended five focus group discussions to talk about how they fed their children and handled food for children and family. The main reasons for stopping breast feeding after 3 to 6 months of birth were that the mothers had to go back to work; some had to stay in the field for a week or more during harvest time. Mothers who did not stay in the field overnight also did not breast feed their child, because they were not aware about the advantage of breast feeding or simply followed the traditional weaning practice.

### 2.2. Mycotoxins in Food Samples

Aflatoxin B_1_ was found in 87.5% of composite food samples except the tofu products group (Table 3). The highest contamination was detected in egg and milk products (5326 ng/kg), followed by oily seed (4086 ng/kg), then meat and meat products (4077 ng/kg) (Table 3). In rice, the aflatoxin B_1_ concentration was 2998 ng/kg. Rice products were consumed in large amounts (Table 2). There have been a few surveys of mycotoxins in foods in Vietnam, including small sample sizes; however, they indicated that aflatoxins are common in maize kernel and maize flour [21,22]. In Lao Cai, it was reported that 25% of self-supplied cereal samples collected in households were contaminated with aflatoxins [23].

Among 40 composite samples analyzed, ochratoxin A was found in 20 samples (49.5%), with the highest concentration (9683 ng/kg, range 9208–10158 ng/kg) found in bean products. Lower ochratoxin A concentrations were shown in the food groups of animal original such as aquaculture products (4850 ng/kg), egg and milk products (3164; 2930–3402 ng/kg), meat products (2685; 2339–3030 ng/kg) and fish products (2245; 1770–2720 ng/kg) (Table 3). In contrast, the concentration of ochratoxin A in all staple cereal samples (rice products, wheat products, other cereal and tube, roof products) was below the detection limit.

Only one black bean and one milk composite sample were found to be contaminated by fumonisins. Among 25 cereal samples collected in various locations of Vietnam, Trung found that eight samples (32%) were contaminated with fumonisins with concentrations ranging from 400 to 3300 ng/g [22]. We have previously reported fumonisins in 8.1% of rice and 23.5% of maize in households supplying their own cereals in Lao Cai province [23].

### 2.3. Growth Indicators and Their Correlates

The overall proportions of stunted children (HAZ < −2) were 50.4% (60/119), 3.4% (4/119) of the children were classified as wasted (WHZ < −2). Mean HAZ was −1.94 (range: −3.31–2.50), mean WHZ was −0.57 (range: −3.33–3.27). Some of the z-scores are summarized in Table 4 listed by age group and gender together with selected nutritional intake measures and estimated intake of mycotoxins. Differences between boys and girls were minor, while the older age group had lower z-scores than the younger group. A significant difference of vitamin A daily intake (*p* < 0.05) was observed between the two age groups of boys only.

In the principal component analysis (PCA) analysis of the dietary variables, the first seven principal components explained 89.1% of the variation in dietary intake with the 7th component having an eigenvalue of 1.04. The loadings (only those above 0.3 are shown) of the included food intake variables on the seven rotated components are shown in Table 5. Loadings are correlations between the original dietary variables and the principal components. Many variables loaded slightly on the unrotated component 1, which likely represents the amount of food eaten, i.e., carbohydrate, non-animal protein and total energy intake loaded most strongly on the rotated component 1. Zinc intake was another factor loading on component 1 (Table 5). High loadings on components 2 to 5 were mainly various vitamin and mineral variables, while for component 6, vegetable fat/oil, vitamin A from non-animal sources, the fiber content and fat from animals were important (Table 5). On component 7, the most important variables were protein from meat and iron derived from meat (Table 5). The seven principal component scores were used as potential correlates in the growth indicator analyses.

Mycotoxin exposure estimates showed a skewed distribution, and scores were therefore log_n_-transformed. Pairwise correlations between exposures to the three toxins were high, i.e., correlation coefficients varied from 0.85 to 0.98 (results now shown). This could obviously result in problems of collinearity in regression models where the three toxins were used as simultaneous correlates. Hence, we performed a principal component analysis on log_n_(exposure) of the three toxins. The first principal component accounted for 94.3% of the total variation in mycotoxin exposure and all three toxins loaded similarly on the first component. The principal component scores for the first component were used as a correlate in further analysis of correlation between toxins and the growth indicators.

We tested a number of potential correlates of HAZ and WHZ scores one by one (for each score adjusting for age in months and gender) and jointly in multivariable analyses where age, gender and energy consumption was forced into the model (data not shown). The HAZ and WHZ scores declined with increase in age in a linear manner (Table 4). None of the mycotoxins or the combined principal component score was significantly correlated with HAZ or WHZ when tested alone together with age and gender (data not shown).

In the final analysis, we decided to model for each toxin separately and the combined score from PCA. Age and gender were considered potential correlates and were included in any model. We tried four different models (Y = growth indicator and T = toxins, individual or combined; the factors included in brackets were forced into the model): (1) Y = b1×age + b2×gender + b3×T + const.; (2) Y = b1×age + b2×gender + b3×T + b4×Energy + const.; (3) Y = (b1×age + b2×gender + b3×T + b4×Energy) + b5×VitA + b6×Zn + b7×Fe + const; and (4) Y = (b1×age + b2×gender + b3×T + b4×PC1) + b5×PC2 + …. + b10×PC7 + const. Results are summarized in Table 6. None of the toxins were significantly correlated with the growth indicators when adjusting for age and gender. When adjustments were also made for total energy (model 2), all toxins showed a significant correlation with HAZ but not with WHZ. This was also the case when adding vitamin A, total protein, iron and zinc (model 3). When adjusting for dietary intake using the principal component scores, all three toxins showed a negative correlation with HAZ, while only aflatoxin B_1_ and fumonisin were negatively correlated with WHZ (Table 6).

Children such as the ones studied in the Lao Cai province are constantly exposed to numerous mycotoxins in the food chain. There are several studies linking aflatoxin intake to growth impairment in children. A dose-response relationship between high aflatoxin levels in the blood and low WAZ (*p* = 0.005) and HAZ scores (*p* = 0.001) were found in a cross-sectional study in Togo and Benin [24]. A study in the Gambia found an association between high exposure to aflatoxin in utero and low weight (*p* = 0.012) and length gains (*p* = 0.044) in the first year of life [25]. A strong negative correlation between blood aflatoxin levels and child growth (stunting) was reported in a longitudinal study of 200 children between 16 and 37 months of age. Fumonisin exposure was pointed out to be a possible factor in slowed child growth as levels of urinomarker of fumonisin B_1_ concentration were negatively associated with growth [26].

### 2.4. Risk Assessment for Mycotoxin Exposure

#### 2.4.1. Aflatoxin B_1_

Using the data of contamination level and daily intake of each food group, mean dietary exposure of aflatoxin B_1_ was estimated at 118.7 ng/kgbw/day (range 104.9–124.2 ng/kgbw/day) resulting in a risk of hepatocellular carcinomas (HCC) of 12.1 cases/100,000 individual/year (range 10.7–12.7 cases/100,000 individual/year). The rice product group was found to be the main source of aflatoxin B_1_ exposure (52.2 ng/kg bw/day), therefore contributed with the highest risk (5.3 cases/100,000 individual/year) of HCC in comparison to other food groups (Table 7). Our previous study in Lao Cai on risks for HCC when consuming self-supplied staple cereals showed that the dietary exposure to aflatoxins and risk of HCC were 33.7 ng/kg bw/day and 2.7 cases/100,000 individual/year, respectively [23].

In line with the above risk estimation, MOEs of aflatoxin B_1_ of all food groups are far lower than 10,000 (range from 3 to 532), resulting in a combined MOE of total aflatoxin B_1_ daily intake as low as 1.4, which is of major public health concern. This is supported by evidence of increased susceptibility to cancer from early-life exposure, particularly for chemicals acting through a genetoxic mode of action like aflatoxins [1]. The high dietary intake exposure of aflatoxins found in the present study together with HBV and HCV infections is likely to represent increased risks of children to liver cancer much more than for adults.

#### 2.4.2. Ochratoxin A

An amount of 52.6 ng/kg bw/day was estimated as the average ochratoxin A exposure, while 77.0 ng/kg bw/day was the highest exposure value (Table 7). The mean and high dietary exposure levels of ochratoxin A were, respectively, equivalent to 261% and 413% of PTDI (14 ng/kg bw/day) [29]. It should be noted that the ochratoxin A exposure level in our study was based on average food intake of children only, which means that the actual exposure dose with the 95th percentile might be much higher. Among the few reports on exposure of children to ochratoxin A, children aged 4 to 6 years were found to be the age group with the highest ochratoxin exposure in the Czech Republic [30]. Results from a French total diet study showed that the estimated average intake of ochratoxin A in children was 4.1 ng/kg bw/day with the 95th percentile exposure being 7.8 ng/kg bw/day [31]. Ochratoxin A contamination of raw pork and meat products is detected quite commonly in Europe [32,33,34]. Mycotoxins in meat originate mainly from contaminated feed. In our study, the food groups contributing the most to ochratoxin A exposure were rice products (14.2 ng/kg bw/day) followed by egg and milk products (10.7 ng/kg bw/day) and beans (9.8 ng/kg bw/day). Thus, a MOE of less than 10,000 was observed for the five food groups. Taking into account the ochratoxin A exposure level of food groups, MOE of the total daily intake was 400, which represents a real risk for renal cancer in the study population.

#### 2.4.3. Fumonisins

Although fumonisins contamination was the least common of the mycotoxins studied; still, an average and highest exposure dose of 1250 and 1929 ng/kg bw/day, equal to 63% and 96% of PTDI (2000 ng/kg bw/day), respectively, were observed, using a hepato-carcinogen benchmark dose lower limit 10% (BMDL_10_) of 150 µg/kg bw/day [35]. Assuming that the contamination level of fumonisin B_1_ is 70% of that of total determined fumonisins [17], the MOE of fumonisin B_1_ in total daily intake was 1713, indicating a health risk for the children due to consumption of large portions of various food items containing low levels of fumonisins.

#### 2.4.4. Aged Adjusted MOEs of the Mycotoxins

Cancer risk assessment methods currently assume that children and adults are equally susceptible to exposure to chemicals. However, research indicates that individuals exposed to mycotoxins at a young age are at higher risk developing cancer than adults [36]. Consequently, a modifying factor may need to be applied to our cancer-risk estimates to ensure risks are not underestimated. The US EPA calculated age-dependent adjustment factors (ADAFs) to account for that children are more susceptible to carcinogens [28]. These factors, which apply to carcinogens with a genotoxic mode of action, are as follows: ADAF is 10 for children <2 years of age; ADAF is 3.16 for children aged 2 to <16 years; and there should be no adjustment (ADAF = 1) for children ≥16 years of age. The MOEs adjusted by ADFA of aflatoxin B1 and ochratoxin A in total daily intake were calculated and are shown in Table 7, while the one of fumonisin B_1_ was 542.

#### 2.4.5. Combined Exposure to All Three Mycotoxins

Co-occurrence of mycotoxins is common worldwide [37]. A study in Tanzania showed that in three geographically distant villages, 82% (*n* = 148) of children aged 12 to 22 months were exposed to both aflatoxin and fumonisins [26]. Studies in Asian countries show that aflatoxin and fumonisin are commonly found together in foods [37]. In our study, frequency histograms of the mycotoxins showed a skewed distribution and scores were therefore log-transformed. Pairwise correlations between the three toxins were high, i.e., correlation coefficients from 0.8457 to 0.9772 document a frequent co-exposure to the mycotoxins studied. We know too little about the toxicity associated with exposure to multiple mycotoxins, e.g., additive, synergistic or antagonistic toxic effects.

## 3. Conclusions

We estimated exposure to aflatoxin B1, ochratoxin A and fumonisins among children in the Lao Cai province using a total dietary study (TDS) approach. Exposures to all three mycotoxins were high and exceeded toxicological reference levels. Risk assessments showed a high risk for liver cancer due to the consumption of aflatoxin B_1_ contaminated foods and lower risks for liver cancer due to fumonisin exposure and renal cancer due to ochratoxin A exposure. Furthermore, high exposure to mycotoxins was associated with impaired child growth when adjusted for age, gender and dietary intake. Though the mechanisms are not clear, stunning and the associated compromised immunity together with high mycotoxin exposure are likely to further negatively impact child development. Locally adapted post-harvest interventions that effectively reduce mycotoxin development in stable cereals are needed.

## 4. Materials and Methods

### 4.1. Study Area

The study took place in and covered the entire Lao Cai province, which consists of nine sub-regions; Lao Cai city itself, together with eights districts (Figure 1).

### 4.2. Anthropometric Measurement

An anthropometric study was conducted in the Ta Phoi and Hop Thanh communes, Lao Cai district, where the inhabitants represented five ethnic groups, i.e., Dao, Giay, Xapho, Tay and Kinh. From a list of 300 households, all 119 children aged 13 to 59 months were selected. Children were weighed and measured once while wearing light-weight clothing following WHO’s instructions [38]. Children younger than 24 months of age were laid horizontally and weighed using a children scale. Their length was also measured using a measuring tape. Children aged 24 to 59 months were weighed barefoot using an electronic scale. The height of these children was measured using a stadiometer while standing straight on a horizontal surface with their heels together and eyes straight forward.

### 4.3. Daily Food Intake Surveillance

The food consumed by the children studied was estimated based on information collected from 24 h recall food intake interviews conducted on three consecutive days combined with actual weighing of the reported consumptions [39]. The mother or grandmother was interviewed on the types of dishes consumed during the last day, including information about all ingredients used for food preparation. Supporting tools, such as spoon, table spoon, bowl and cups, were used to activate the household member’s memory and to allow subsequent weighing of the foods. Accordingly, available foods were weighted for confirming amount stated by household members using a Tanita electronic scale, (Tokyo, Japan).

For collecting further information about feeding practices of children, five focus group discussions were carried out with mothers or caregivers. Eight to 10 mothers belonging to the same ethical group were invited to discuss about breast feeding, complementary feeding, food safety practice and taking care of sick children.

### 4.4. Mycotoxins Exposure Risk Assessment

The guidance for Total Dietary Study (TSD) approach issued by EFSA, WHO and FAO [40] was employed to assess dietary exposure of the aflatoxin B_1_, ochratoxin A and fumonisins of children younger than 5 years old.

#### 4.4.1. Food Sample Collection and Analysis

Data collected in the daily food intake surveillance came up with a list containing 89 food items. Of these, 40 were selected for the TDS (Table 1). The selection was made on the basis that these food items were most commonly consumed and probably could be contaminated with one or more of the three mycotoxins analyzed. In each of the nine sub-regions, three retail markets were selected. Choosing one retailer at each market made up a total of 27 retailers. At each of the 27 retailers, three independent samples of about 100 g size were collected for each of the 40 pre-selected food items. Thus, 1080 individual foods samples were collected (40 food items × 9 sub-regions × 3 retail markets).

For each of the 40 composite food items, samples taken were compounded in the following way. The three 100 g samples from a given retailer were mixed, and from the 300 g of the resulting mixed sample a 100 g sub-sample were taken. The nine sub-samples of 100 g, representing each of the nine sub-regions, were then mixed, to give a sample of 900 g representing the province. Three hundred grams of this sample was taken for preparation and cooking according to the most common local cooking practices. The means of preparation and cooking complied with the EFSA/FAO/WHO guidance in kitchen preparation [40]. In total, this procedure resulted in 40 composite cooked samples each representing one food item as “averaged” over the whole of the province. Each of these 40 food item samples were analyzed for the three mycotoxins mentioned above as describe in the following.

#### 4.4.2. Mycotoxin Contamination Analysis

ELISA-based methods with aflatoxin B_1_, ochratoxin A and fumonisin B_1_ as standards and commercially available detection kits (AgraQuant^®^, Romer Labs, Inc., Newark, DE, USA) were used for aflatoxin B_1_ (COKAQ 8000, limit of detection is 2 ng/g), ochratoxin A (COKAQ 2000, limit of detection is 1.9 ng/g), and fumonisins (COKAQ 3000, limit of detection is 0.2 µg/g) analyses according to the manufacturer’s instructions and as reported previously [23]. Briefly, for each sample, one extract was produced then duplicate determinations of the toxin were performed. Standard curves were plotted using standard aflatoxin B_1_, ochratoxin A and fumonisin B_1_. The concentration of aflatoxin B_1_, ochratoxin A and fumonisins were calculated on a dry weight basis according to the specifications of the manufacturer. The sample moisture content was measured by drying 10.0 g in an oven at 105 °C for 17 h [41].

#### 4.4.3. Mycotoxins Exposure

The deterministic (or single point) approach was adopted to estimate the dietary mycotoxin exposure [39]. According to these recommendations, half of the limit of detection (LOD) was used for all results of aflatoxin B_1_ less than LOD, since concentration of the mycotoxin was below LOD in less than 60% of samples. In contrast, since the contamination level of ochratoxin A and fumonisins in more than 60% of samples were lower than LOD, then two estimates using zero (lower bound) and LOD (upper bound) for all results less than LOD were applied.

The chronic daily exposure to each of the mycotoxins was calculated based on the mycotoxin contamination level of each TDS food group (ng/kg food) and the daily intake (kg food/day) of this food group using an 11.3 kg mean of body weight of children.

#### 4.4.4. Risk Characterization

As recommended by EFSA and JEFCA, the MOEs of all three mycotoxins were calculated [18,19]. The MOE was given by the ratio between the benchmark dose level that caused a 10% increase in cancer incidence in animal (BMDL _10_) and the total intake (MOE = BMDL_10_/total intake) [18]. For estimation of MOE, BMLD 10 of developing hepatocellular carcinoma HCC (170 ng/kg bw/day and 150 µg/kg bw/day; 95% lower confidence limit) were applied for aflatoxin B_1_ [6] and fumonisins [35], respectively. For ochratoxin A, a MOE based on the lowest BMDL10 associated with an increase in renal cancer (21 µg/kg bw/day) by exposure to ochratoxin A was determined [27]. MOE values lower than 10,000 may indicate a public health concern [18].

The mycotoxin of most concern is aflatoxin B_1_, which has been reported to increase liver cancer among people infected with hepatitis virus. Risk assessment for aflatoxin B_1_ was performed based on the dietary exposure to aflatoxin B_1_ and its potency using the prevalence of individuals being hepatitis B surface antigen- (HbsAg) positive and having a primary liver cancer potency of 0.3 cancers per year per 100,000 population per ng aflatoxin B_1_/kg body weight (kg bw)/day and the negative individuals set to have 0.01 cancers per year per 100,000 population per ng AFB_1_/kg bw/day [12,42]. In this study, we assumed that 1% of children younger than five years old were HbsAg-positive [14].

### 4.5. Data Analysis

The WHO standards were used to determine the nutritional status of children, i.e., weight for age (WAZ), height (length) for age (HAZ or LAZ) and weight for height (WHZ) z-scores [20]. Descriptive statistics for these growth indicators, selected variables on dietary intake and mycotoxin exposure were summarized by gender and age group (13–23 months versus 24–59 months). The two age groups, however, were represented with very different sample sizes and in subsequent analyses age in months was used. Linearity of the associations between various outcome variables and age in month was tested by polynomial regression [43].

The dietary variables could all be potential correlates of the growth indicators. Correlation coefficients between pairs of dietary variables ranged from −0.14 to 0.94. The 50, 75 and 90 percentile of all possible pairwise correlations among the dietary variables (*n* = 528) were 0.38, 0.61 and 0.76, respectively. Therefore, we conducted a Principal Component Analysis (PCA) on these variables. The number of components retained was based on a scree plot, the retained components were then submitted to a varimax rotation [44] and the factor scores were used as predictors in the regression analysis.

Correlates of the growth indicators were tested using multiple linear regression, where age (months) and gender were entered as well. Regression models were either specified by us or using a backwards stepwise regression procedure (*p* for removal = 0.051; *p* for entry = 0.050), but with some factors (see results) forced into the model. Mycotoxin species were tested individually as correlates of the growth indicators, and since these toxins were all correlated, they were also tested as a combined score based on the principal component score.

### 4.6. Ethical clearance

Mother or caregivers of all 119 subjects gave their informed consent for their attendance before they participated in the study. The study was conducted in accordance with the Declaration of Helsinki, and the protocol was approved by the Ethics Committee of the National Institute of Nutrition, Hanoi, Vietnam (ID 6 VDD 2009, dated 8 September 2009).

## Figures and Tables

**Figure 1 toxins-11-00638-f001:**
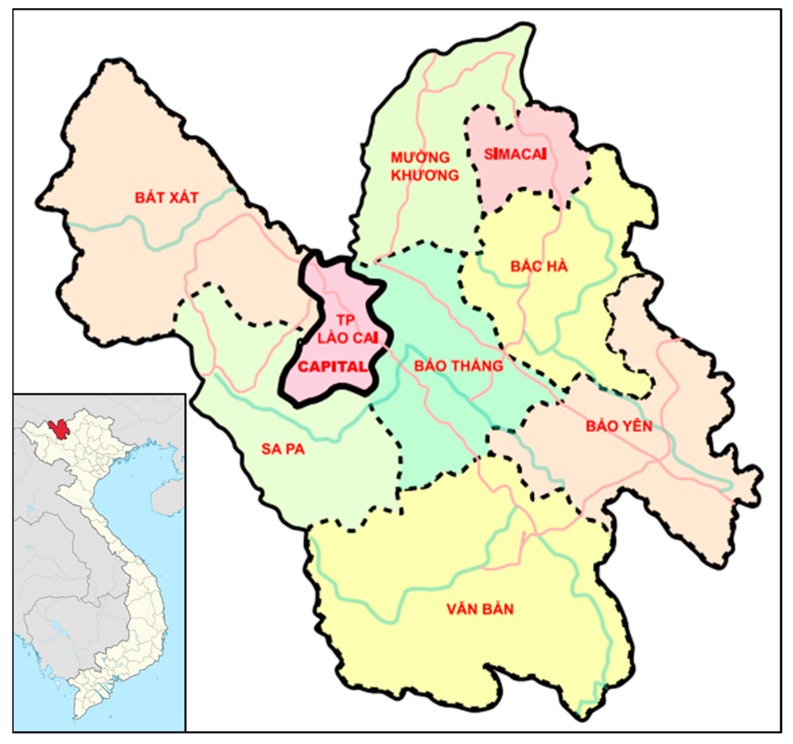
Map of nine districts of the Lao Cai province.

**Table 1 toxins-11-00638-t001:** Food groups and food preparation procedures in households in Lao Cai province, Vietnam.

Food Groups ^a^		Food Items	Food Preparation ^b^
1	Rice and products	Rice	Boiled
		Sticky rice	Boiled
		Rice noodle	Boiled
2	Wheat and products	Noodle	Boiled
3	Tubes, root and products	Vicermine	Boiled
		Shrimp chip	Deep fried
4	Beans and products	Black bean	Stewed
		Mung bean	Stewed
		Soybean milk	Ready to eat
		Soy bean	Stewed
5	Tofu	Tofu	Boiled
6	Oily seeds	Peanut	Stir fried
7	Vegetables	Bamboo shot, fermented	Boiled
8	Sugar, confectionary	Biscuit	Ready to eat
		Wafers	Ready to eat
		Cookies	Ready to eat
		Sesame candy	Ready to eat
		Nugget/peanut candy	Ready to eat
9	Oil, fat	Pork, fat	Fried
		Cooking oil	
10	Meat and products	Dry pork meat	Ready to eat
		Pork pie, fried	Ready to eat
		Pork pie, boiled	Ready to eat
		Pork rib, boneless	Stewed
		Pigeon	Stewed
		Beef	Stir fried
		Dog meat	Boiled
		Chicken	Boiled
		Pork, lean	Boiled, stir fried
		Pork	Boiled, stir fried
		Pork liver	Stir fried
11	Egg and milk	Egg, chicken	Boiled, fried
		Egg, duck	Boiled, fried
		Condensed Milk	Ready to eat
		Milk powder	Ready to eat
		Milk	Ready to eat
12	Fish	Dried fish	Stir fried
		Fish, fresh water	Boiled
13	Other aquaculture products	Dried shrimp	Boiled
		Shrimp	Stir fried

^a^ Food groups were categorized according to a previous national survey [16]. ^b^ Food items were prepared as practiced by households in the Lao Cai province.

**Table 2 toxins-11-00638-t002:** Food and nutrient intake amongst children in Lao Cai.

	Mean	Range
**Food Intake (g per child per day)**		
Rice and products	196	49–313
Wheat and products	11	0–93
Tubes, root and products	3	0–100
Bean and products	11	0–293
Tofu	4	0–63
Oily seed	2	0–29
Vegetable leaf	50	0–149
Vegetable tube	12	0–157
Fruit	22	0–225
Confectionary	15	0–215
Seasoning	0	0–4
Oil, fat	2	0–12
Meat and products	30	0–110
Egg and milk	38	0–281
Fish	6	0–31
Other aquaculture products	2	0–55
Other spices	0	0–4
**Dietary Composition (per child per day)**		
Energy (kcal)	871	218–171
Protein (g) total	28	8–67
Protein from animal sources (g)	10	0–50
Non-animal protein (g)	18	4–33
Protein (eggs and milk) (g)	3	0–17
Protein from meat (g)	6	0–45
Carbohydrate (g)	152	37–258
Fat (g)	17	2–57
Vegetable fat/oil (g)	7	1–48
Fiber (g)	2.5	0.4–6.9
Ash (g)	3.5	0.9–7.2
Total vitamin A ^a^ (mcg)	99.0	0–1044.0
Animal source vitamin A ^a^ (mcg)	90.0	0–1044.0
Non-animal vitamin A ^a^ (mcg)	9.0	0–145.0
Carotenoid (mcg)	2353.0	0–8576.0
Vitamin C (mg)	33.1	0.0–170.3
Thiamin (mg)	0.4	0.1–1.0
Riboflavin (mg)	0.3	0.0–1.1
Niacin (mg)	5.1	1.2–13.2
Vitamin D (mcg)	0.4	0.0–4.7
Folic acid (mcg)	0.0	0.0–0.0
Folate (mcg)	94.5	8.6–308.3
Vitamin B12 (mcg)	0.6	0.0–4.2
Calcium (mg)	181.8	27.8–707.9
Sodium (mg)	167	8–1087
Potassium (mg)	784.9	176.1–1716.6
Magnesium (mg)	69.5	11.7–177.4
Zinc (mg)	3.7	1.1–6.7
Phosphorous (mg)	361	73–905
Iron (mg)	4.8	1.0–9.5
Iron from meat/fish/poultry (mg)	0.5	0.0–3.4

^a^ Retinol equivalent.

**Table 3 toxins-11-00638-t003:** Aflatoxin B_1_ and ochratoxin A contents (ng/kg) in food groups included in the total dietary study.

Food Group ^a^	Number of Composite Samples	Aflatoxin B_1_			Ochratoxin A		
		Number of Test Results < LOD	Concentration (ng/kg) ^b^		Number of Test Results < LOD	Concentration (ng/kg) ^b^	
			MB	LB-UB		MB	LB-UB
Rice and products	3	1	2989	2400–3020	3	950	0–1900
Wheat and products	1	0	1000	1000	1	950	0–1900
Tubes, root and products	2	1	2171	1670–2670	2	950	0–190
Beans and products	4	1	2864	2610–3110	2	9683	9210–10,160
Tofu	1	1	1000	0–2000	1	950	0–1900
Oily seeds	1	0	4086	4086	1	950	0–1900
Vegetables	1	0	3470	3470	1	950	0–1900
Sugar/confectionary	5	0	4033	4033	4	1173	410–1930
Oil, fat	2	0	3382	3382	1	1462	980–1940
Meat and products	11	1	4077	3990–4170	4	2685	2340–3032
Egg and milk	4	0	5326	5325	1	3164	2930–3400
Fish	2	1	2301	1800–2800	1	2245	1770–2720
Other aquaculture products	3	0	2518	1850–3180	0	4850	4850
Total	40	6			20		

^a^ Food groups were categorized according to a previous national survey [16]. ^b^ Medium bound (MB) figures (ND = LOD/2) were used as mean values. Lower bound (LB) and upper bound (UB) figures. LOD, limit of detection.

**Table 4 toxins-11-00638-t004:** Anthropometric measurements, selected dietary intake and mycotoxin exposure (mean and range) by age group and gender.

		Boy		Girl		*p*-Value
		*n*	Mean and Range	*n*	Mean and Range	
**Anthropometric Measurement**						
Length/height for age Z- score	13–23 months	6	−0.34 (−0.76–0.65)	8	1.01 (−2.36–2.50)	n.s.
	24–59 months	58	−2.22 (−3.19–1.52)	47	−2.29 (−3.31–1.60)	n.s.
	*p*-value		<0.01		<0.001	
Weight for length/height Z- score	13–23 months	6	−0.49 (−1.05–0.37)	8	0.27 (−1.00–2.00)	n.s.
	24–59 months	58	−0.66 (−2.33–1.13)	47	−0.61 (−2.41–3.27)	n.s.
	*p*-value		n.s.		<0.05	
% Length/height for age Z- score < −2 (%)	13–23 months	6	0	8	12.5	n.s.
	24–59 months	58	53.4	47	59.6	n.s.
	*p*-value		<0.05		<0.05	
**Dietary Intake**						
Energy intake(kcal/day)	13–23 months	6	790 (434–1097)	9	742 (367–1164)	n.s.
	24–59 months	58	901 (218–1436)	47	868 (378–1713)	n.s.
	*p*-value		n.s.		n.s.	
Protein intake (g/day)	13–23 months	6	24 (11–49)	9	22 (9–38)	n.s.
	24-59 months	58	29 (8–67)	47	28 (9–48)	n.s.
	*p*-value		n.s.		n.s.	
Vitamin A intake (mcg/day)	13–23 months	6	15.5 (0.0–60.9)	9	47.1 (0.0–160.9)	n.s.
	24–59 months	58	95.3 (0.0–629.8)	47	124.1 (0.0–1043.7)	n.s.
	*p*-value		<0.05		n.s.	
Iron intake (mg/day)	13–23 months	6	5.6 (3.7–8.3)	9	4.6 (1.9–6.5)	n.s.
	24–59 months	58	4.8 (1.1–9.5)	47	4.7 (1.6–9.4)	n.s.
	*p*-value		n.s.		n.s.	
Zinc intake (mg/day)	13–23 months	6	3.7 (2.6–5.7)	9	3.6 (1.1–4.8)	n.s.
	24-59 months	58	3.8 (1.1–6.7)	47	3.75 (1.5–6.5)	n.s.
	*p*-value		n.s.		n.s.	
**Mycotoxin Exposure**						
Aflatoxin B1 (ng/kg bw/day)	13–23 months	6	135.9 (87.2–170.3)	8	100.5 (49.1–156.6)	n.s.
	24–59 months	58	123.5 (28.4–247.3)	47	121.6 (40.2–246.3)	n.s.
	*p*-value		n.s.		n.s.	
Fumonisins (ng/kg bw/day)	13–23 months	6	3.6 (2.1–4.6)	8	2.7 (1.6–4.0)	n.s.
	24–59 months	58	3.5 (0.8–7.5)	47	3.5 (1.3–7.1)	n.s.
	*p*-value		n.s.		n.s.	
Ochratoxin A (ng/kg bw/day)	13–23 months	6	43.2 (20.4–82.1)	8	31.3 (17.6–47.2)	n.s.
	24–59 months	58	54.8 (11.0–344.7)	47	57.2 (13.7–239.5)	n.s.
	*p*-value		n.s.		n.s.	

n.s.: not significant.

**Table 5 toxins-11-00638-t005:** Correlations (loadings) between dietary variables and the rotated principal components (Comp 1 to 7). Only loadings above 0.3 are shown. Factors not loading on the first seven components are not shown.

Variable Label	Principal Component Score						
	Comp 1	Comp 2	Comp 3	Comp 4	Comp 5	Comp 6	Comp 7
Energy (Kcal)	0.38						
Non-animal protein sources (g)	0.39						
Carbohydrate by difference (g)	0.51						
Zinc (mg)	0.33						
Riboflavin (mg)		0.31					
Vitamin D (mcg)		0.44					
Calcium (mg)		0.38					
Sodium (mg)		0.40					
Poly-unsaturated fatty acid (g)			0.50				
Mono- saturated fatty acid (g)			0.62				
Animal source vitamin A (mcg)				0.55			
Vitamin B12 (mcg)				0.48			
Cholesterol (g)				0.50			
Carotenoid (mcg)					0.58		
Vitamin C (mg)					0.44		
Folate (mcg)					0.48		
Vegetable Fat/oil (g)						0.57	
Fiber (dietary fiber) (g)						0.32	
Fat (g)						0.33	
Non-animal source vitamin A						0.57	
Protein from meat (mg)							0.58
Niacin (mg)							0.31
Iron from fish, poultry and other meat product (mg)							0.64

**Table 6 toxins-11-00638-t006:** Multivariable analyses of potential correlates of HAZ and WHZ tested adjusting for age and gender using four different models.

Model	Factors Adjusted for (Forced into Model)	Other Potential Correlates	Log_n_ (Aflatoxin B_1_ Exposure)	Log_n_ (Fuminosin Exposure)	Log_n_ (Ochratoxin A Exposure)	Combined (Based on PCA Score)
**HAZ**						
1	Age (months) + gender	None	0.21 (−0.40–0.81)	0.11 (−0.54–0.75)	−0.07 (−0.48–0.35)	0.02 (−0.13–0.17)
2	Age (months) + gender + total energy	None	−1.13 (−1.81–−0.45) **	−1.52 (−2.24–−0.80) ***	−0.76 (−1.18–−0.35) ***	−0.32 (−0.49–−0.16) ***
3	Age (months) + gender + total energy	Vitamin A; total protein; iron; zinc	−2.19 (−2.80–−1.58) ***	−2.62 (−3.24–−1.99) ***	−1.24 (−1.62–−0.86) ***	−0.58 (−0.72–−0.43) ***
4	Age (months) + gender + PC1	PC2 to PC7 ^a^	−2.66 (−3.40–−1.92) ***	−2.99 (−3.71–−2.27) ***	−0.96 (−1.31–−0.61) ***	−0.66 (−0.83–−0.48) ***
**WHZ**						
1	Age (months) + gender	None	−0.16 (−0.56–0.25)	−0.14 (−0.57–0.28)	0.01 (−0.26–0.29)	−0.02 (−0.12–0.08)
2	Age (months) + gender + total energy	None	−0.26 (−0.78–0.26)	−0.26 (−0.82–0.30)	0.0145 (−0.30–0.33)	−0.04 (−0.17–0.09)
3	Age (months) + gender + total energy	Vitamin A; total protein; iron; zinc	−0.534 (−1.07–0.00) *	−0.50 (−1.07–0.084)	−0.08 (−0.40–0.24)	−0.09 (−0.22–0.04)
4	Age (months) + gender + PC1	PC2 to PC7	−1.50 (−2.17–−0.83) ***	−1.26 (−1.95–−0.56) ***	−0.41 (−0.81–0.00)	−0.27 (−0.44–−0.11) **

* *p*-value < 0.05, ** *p*-value < 0.01, *** *p*-value < 0.001. ^a^ Principal component scores (PC1-PC7) were used as potential correlates of growth indicators. PC1 is a measure of total food intake.

**Table 7 toxins-11-00638-t007:** Dietary exposure to aflatoxin B_1_ and ochratoxin A and risk of liver and renal cancer.

Food Groups ^a^	Aflatoxin B_1_						Ochratoxin A			
	Exposure (ng/kg bw ^b^/day)		HCC Risk ^d^ (cases/100,000 Population)		MOE^e^_HCC_	MOE^f^_HCC_ Adjusted	Exposure (ng/kg bw/day)		MOE^e^ _RC_	MOE^f^_RC_ Adjusted
	MB^c^	LB-UB^c^	MB	LB-UB			MB	LB-UB		
Rice and products	52.2	41.2–52.8	5.3	4.2–5.4	3	1	14.2	0–1900	1478	468
Wheat and products	1.0	0–1.9	0.1	0–0.2	183	58	0.9	0–1900	>10,000	7384
Tubes, roof and products	0.7	0.5–0.8	0.1	0–0.1	261	83	0.3	0–1900	>10,000	>10,000
Beans and products	3.9	3.6–4.1	0.4	0.4	44	14	9.8	9208–10,158	2142	678
Tofu	0.3	0–0.6	0.0	0–0.1	532	168	0.3	0–1900	>10,000	>10,000
Oily seeds	0.6	0.6	0.1	0.1	287	91	0.1	0–1900	>10,000	>10,000
Vegetables	18.9	18.9	2.0	2	9	3	5.2	0–1900	4,038	1278
Sugar/confectionary	6.8	5.5	0.7	0.7	25	10	1.6	413–1933	>10,000	4153
Oil, fat	0.5	0.5	0.1	0.1	347	110	0.2	987–1937	>10,000	>10,000
Meat and products	13.8	13.5–14.0	1.4	1.4	12	4	7.1	1339–3030	2957	936
Egg and milk	18.0	18	1.9	1.9	9	3	10.7	2927–3401	1962	621
Fish	1.3	1.0–1.6	0.1	0.2	133	42	1.2	1770–2720	>10,000	5538
Other aquaculture products	0.4	0.3–0.6	0.0	0–0.1	384	122	0.9	4850	>10,000	7384
Total	118.7	104.9–124.2	12.1	10.7–12.7	1.4	0.5	52.6	29.7–77.0	400	127

^a^ Food groups were categorized followed those applied in data of National Survey in the year 2010 [16]. ^b^ Mean body weight (bw) of children was 11.3 kg. ^c^ Medium bound (MB) figures (ND = LOD/2) were used for mean. Lower bound (LB) and upper bound (UB) figures (ND = 0, ND = LOD) were used for range. ^d^ Children risk of hepatitis carcinogen is calculated on the assumption of HbsAg + prevalence 2% [14] and mean exposure. ^e^ MOE, Margin of exposure, based on the calculated as a ratio of benchmark dose lower limit 10% lower bound of AFB_1_ (170 ng/kw bw/day [6]) or OTA (21 µg/kg bw/day [27]) and MB of exposure. HCC, hepatocellular cancer; RC, renal cancer. ^f^ MOEs adjusted by age-dependent adjustment factors for children aged 2–16 (ADAF = 3.16) [28].

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
