# Peer review of "Total Dietary Intake and Health Risks Associated with Exposure to Aflatoxin B1, Ochratoxin A and Fuminisins of Children in Lao Cai Province, Vietnam"

_toxins, 2019, doi:10.3390/toxins11110638_

Round 1

Reviewer 1 Report

Given the health and economic burden associated with Aflatoxin B1, Ochratoxin A and 4 Fuminisins contamination of food, the manuscript deals with a topic of high interest.

In my opinion, it is recommended for publication after some minor corrections:

A few punctuation problems are present in the manuscript. I suggest the Authors to double-check the text.

A dedicated paragraph on statistical analysis is recommended in the Materials and Methods section.

A PCA chart can be a good way to visualize data.

Author Response

Given the health and economic burden associated with Aflatoxin B1, Ochratoxin A and  Fuminisins contamination of food, the manuscript deals with a topic of high interest.

Our response: We are happy to learn that the reviewer finds that we address an important food safety and health problem.

1 A few punctuation problems are present in the manuscript. I suggest the Authors to double-check the text.

Our response: We have revised the text and done minor corrections as proposed by the reviewer.

2 A dedicated paragraph on statistical analysis is recommended in the Materials and Methods section.

Our response: We are a bit puzzled about this suggestion as there already is a dedicated paragraph on statistical analysis in the Materials and Methods and as far as we can judge, it describes the analytic approach clearly. If the reviewer could be specific as to how the section should be improved, we would be glad to accommodate such suggestions.

A PCA chart can be a good way to visualize data.

Our response: Generally, we agree that PCA charts often are a good way to visualize data. However, since we are dealing with 7 principal components, it is difficult graphing the data because it cannot really be decided which pair of components should be shown. Loadings plots would just reflect loadings as shown in the table. Biplots might be relevant, but again which dimensions should be plotted. In addition, because of the many variables graphs would be congested with labels, especially if we were to show all combinations as a matrix graph. We therefore think that in this case PCA type of graphs would not add value to the interpretation.

Reviewer 2 Report

The paper entitled: “Total Dietary Intake and Health Risks Associated with Exposure to Aflatoxin B1, Ochratoxin A and Fuminisins of Children in Lao Cai Province, VietNam” present a very interesting study concerning the dietary exposure and health risk caused by mycotoxins for children under 5 years living in the Lao Cai province in northern Vietnam. The paper is well written and the subject is of a great concern as dietary exposure to mycotoxins is associated with severe health disorders. The English language is appropriated and the paper could be accepted for publication in Toxins journal after some minor revisions.

Comments

The choice of the assessment of the dietary intake and health risk for AFB1, OTA and FB should be better argued in the introduction as the last occurrence study  published by BIOMIN in 2018 has shown that in SouthEst Asia the most occurring mycotoxins were FB (81%) followed by DON (68%), AFB1 (54%) and ZEN (51%), while only 30% of samples were contaminated with OTA. Table 2 is hard to follow and it should be better split in two (food intake and dietary composition respectively) Table 7 presents the risk assessment for the exposure to AFB1 and OTA, but not for FB1 as stated in the text (pg 12 line 249) M&M, 4.2 Anthropometric measurement. The number of the children used in the study should be mentioned.

Author Response

The paper entitled: “Total Dietary Intake and Health Risks Associated with Exposure to Aflatoxin B1, Ochratoxin A and Fuminisins of Children in Lao Cai Province, VietNam” present a very interesting study concerning the dietary exposure and health risk caused by mycotoxins for children under 5 years living in the Lao Cai province in northern Vietnam. The paper is well written and the subject is of a great concern as dietary exposure to mycotoxins is associated with severe health disorders. The English language is appropriated and the paper could be accepted for publication in Toxins journal after some minor revisions.

Our response: We are happy to learn that the reviewer considered our study is overall very interesting and well written. We did some minor correction as mentioned by the reviewer.

The choice of the assessment of the dietary intake and health risk for AFB1, OTA and FB should be better argued in the introduction as the last occurrence study published by BIOMIN in 2018 has shown that in SouthEst Asia the most occurring mycotoxins were FB (81%) followed by DON (68%), AFB1 (54%) and ZEN (51%), while only 30% of samples were contaminated with OTA.

Our response: We agree with reviewer´s comment and have therefore rephrased a  sentence to argue the reason of selecting aflatoxin B1, ochratoxin A and fumonisin in our study.

Table 2 is hard to follow and it should be better split in two (food intake and dietary composition respectively).

Our response: As suggested, we have re-arranged the data in Table 2 into two main sections illustrating food intake and dietary composition.

Table 7 presents the risk assessment for the exposure to AFB1 and OTA, but not for FB1 as stated in the text (pg 12 line 249) M&M.

Our response: We agree with the reviewer. The paragraph mentioned discusses on the age adjusted MOE of all three type of mycotoxins not only fumonisins. Hence, we separated it into a new section (2.4.4. Aged adjusted MOEs of the mycotoxins).

Anthropometric measurement. The number of the children used in the study should be mentioned. 

Our response: We agree and have added the number of children.